# *Polygonum cuspidatum* Extract (Pc-Ex) Containing Emodin Suppresses Lung Cancer-Induced Cachexia by Suppressing TCF4/TWIST1 Complex-Induced PTHrP Expression

**DOI:** 10.3390/nu14071508

**Published:** 2022-04-05

**Authors:** Xue-Quan Fang, Young-Seon Kim, Yoon-Mi Lee, Mingyu Lee, Woo-Jin Lim, Woo-Jong Yim, Min-Woo Han, Ji-Hong Lim

**Affiliations:** 1Department of Biomedical Chemistry, College of Biomedical & Health Science, Konkuk University, 268 Chungwon-daero, Chungju 27478, Korea; gkrrnjs654852@kku.ac.kr (X.-Q.F.); yoonmilee@kku.ac.kr (Y.-M.L.); lwj0908@kku.ac.kr (W.-J.L.); 2Department of Applied Life Science, Graduate School, BK21 Program, Konkuk University, 268 Chungwon-daero, Chungju 27478, Korea; 3Jung-Ang Microbe Research Institute (JM), 398, Jikji-daero, Heungdeok-gu, Cheongju 28576, Korea; yskim0801@kku.ac.kr (Y.-S.K.); ywj0808@naver.com (W.-J.Y.); 4Diabetes and Bio-Research Center, Konkuk University, 268 Chungwon-daero, Chungju 27478, Korea; 5Division of Allergy and Clinical Immunology, Brigham and Women’s Hospital and Department of Medicine, Harvard Medical School, Boston, MA 02115, USA; leemk08@gmail.com; 6Korea Institute of Knowledge Service, 20 Wolpyeongsaetteum-ro, Seo-gu, Daejeon 35226, Korea; hmw0416@gmail.com

**Keywords:** cachexia, emodin, lung cancer, PTHrP, TCF4, TWIST1

## Abstract

Cachexia, which is characterised by the wasting of fat and skeletal muscles, is the most common risk factor for increased mortality rates among patients with advanced lung cancer. *PTHLH* (parathyroid hormone-like hormone) is reported to be involved in the pathogenesis of cancer cachexia. However, the molecular mechanisms underlying the regulation of *PTHLH* expression and the inhibitors of PTHLH have not yet been identified. The *PTHLH* mRNA levels were measured using quantitative real-time polymerase chain reaction, while the PTHrP (parathyroid hormone-related protein) expression levels were measured using Western blotting and enzyme-linked immunosorbent assay. The interaction between TCF4 (Transcription Factor 4) and TWIST1 and the binding of the TCF4–TWIST1 complex to the *PTHLH* promoter were analysed using co-immunoprecipitation and chromatin immunoprecipitation. The results of the mammalian two-hybrid luciferase assay revealed that emodin inhibited TCF4–TWIST1 interaction. The effects of *Polygonum cuspidatum* extract (Pc-Ex), which contains emodin, on cachexia were investigated in vivo using A549 tumour-bearing mice. Ectopic expression of TCF4 upregulated *PTHLH* expression. Conversely, *TCF4* knockdown downregulated *PTHLH* expression in lung cancer cells. The expression of *PTHLH* was upregulated in cells ectopically co-expressing TCF4 and TWIST1 when compared with that in cells expressing TCF4 or TWIST1 alone. Emodin inhibited the interaction between TCF4 and TWIST1 and consequently suppressed the TCF4/TWIST1 complex-induced upregulated mRNA and protein levels of PTHLH and PTHrP. Meanwhile, emodin-containing Pc-Ex significantly alleviated skeletal muscle atrophy and downregulated fat browning-related genes in A549 tumour-bearing mice. Emodin-containing Pc-Ex exerted therapeutic effects on lung cancer-associated cachexia by inhibiting TCF4/TWIST1 complex-induced PTHrP expression.

## 1. Introduction

Lung cancer is the most lethal cancer type threatening human life and health. Approximately 22% of lung cancer-related deaths are attributed to cancer cachexia [1]. Cancer cachexia, which is a metabolic syndrome characterised by loss of body weight and decreased skeletal muscle mass, food intake (anorexia), and white adipose tissue (WAT) weight, decreases the quality of life in 60–80% of patients with cancer [2]. The wasting of skeletal muscles, a symptom of cancer cachexia, results from the dysregulation of protein synthesis and degradation in the skeletal muscles [3]. The expression levels of proteolysis-related genes such as *Trim63* and *Fbxo32* (also known as *Atrogin1*) are upregulated in the lung cancer-induced cachectic muscle [4]. Metabolic imbalances in the skeletal muscle are mediated by tumour-derived pro-inflammatory cytokines, such as tumour necrosis factor-α (TNF-α), interleukin-1 (IL-1), IL-6, and transforming growth factor β1 (TGFβ1) [5]. These cytokines are reported to activate proteolysis and suppress protein synthesis through the activation of nuclear factor-κB (NF-κB), JAK/STAT, and Smad3 in the skeletal muscles [6,7]. Consistently, the serum levels of pro-inflammatory cytokines in patients with cachectic cancer are upregulated when compared with those in weight-stable counterparts [8]. A common feature of cancer cachexia is the loss of fat mass, which can be attributed to decreased lipid storage and lipogenesis and increased lipolysis and thermogenesis in WAT [9]. In patients with advanced cancer, WAT is converted into brown adipose tissue (BAT), which has large numbers of mitochondria and exhibits upregulated levels of thermogenic uncoupling proteins. Fat browning contributes to high energy expenditure in cancer cachexia [10]. Several studies have demonstrated that the monoclonal antibody-mediated neutralisation of IL-1 or IL-6 ameliorates cancer cachexia characterised by loss of skeletal muscle mass and WAT browning [9]. Additionally, the pharmacological inhibition of β3-adrenergic signalling is reported to decrease WAT browning and ameliorate cancer cachexia severity [11]. Many efforts have been made to evaluate biomarkers of cancer cachexia. Metabolic alterations such as amino acids and lipoproteins were occurred in the early stage of cancer cachexia, and cytokines were also altered in the cachexia stage [12,13]. Several studies have examined the therapeutic strategies for cancer cachexia by targeting biomarkers of cancer cachexia. However, therapeutics for the clinical treatment of patients with cancer cachexia are not currently available.

PTHLH, which is encoded by parathyroid hormone-related protein (PTHrP), serves as an external stimulus to maintain bone turnover and skeletal homeostasis [14]. Previous studies have reported that PTHrP promotes the growth, metastasis, and chemoresistance of various cancers, such as lung, colorectal, prostate, pancreatic, and breast cancers [15,16,17,18,19,20]. A recent study reported that tumour-derived PTHrP causes cancer cachexia by promoting WAT browning and skeletal muscle loss [4]. Activated Smad3 promotes the transcription of *PTHLH* in response to TGFβ1, which is also known as a cachexia-inducing factor, in breast cancer [7,17]. Monoclonal antibodies against PTHrP are reported to exert therapeutic effects on breast cancer bone metastasis and lung cancer-driven cachexia [4,15]. These findings suggest that PTHrP is a druggable target for cachexia, as well as for bone metastasis in patients with cancer. However, the mechanism underlying the regulation of PTHrP expression has not been completely elucidated.

The E-protein TCF4 (also called E2-2 and ITF2), which is a class I basic helix-loop-helix (bHLH) transcription factor, is involved in the pathogenesis of Fuchs’ endothelial corneal dystrophy, primary sclerosing cholangitis, Pitt–Hopkins syndrome, and several types of malignant tumours [21]. TCF4 exerts oncogenic effects by promoting the proliferation, invasion, and chemoresistance of melanoma, lung, colorectal, and breast cancers [16,22,23,24,25,26]. In contrast, several studies have demonstrated that TCF4 suppresses the growth and progression, which are associated with poor prognosis, of non-small cell lung carcinoma and colorectal cancer [27,28,29,30]. TWIST1, a bHLH-domain-containing transcription factor, is reported to promote epithelial-to-mesenchymal transition (EMT), which increases the invasiveness and metastasis of epithelial cancers [31]. TWIST1 undergoes homodimerization or heterodimerization with various E-proteins, including TCF3, TCF4, and TCF12 [32]. Additionally, TCF4 interacts with TWIST1, promotes EMT and TGFβ1 signalling, and subsequently enhances bone metastasis in Kras^G12V^-driven lung cancer [16]. The mechanisms underlying the interaction between TCF4 and TWIST1 and the role of the TCF4–TWIST1 complex in cancer cachexia have not been investigated.

This study demonstrated the molecular mechanisms underlying the TCF4/TWIST1 interaction-mediated PTHrP expression in lung cancer-induced cachexia.

## 2. Materials and Methods

### 2.1. Quantitative Real-Time PCR (qRT-PCR)

To measure mRNA expression, total RNA in cultured lung cancer cells was prepared by using TRIzol (Invitrogen, Carlsbad, CA, USA) and isopropanol (Sigma-Aldrich, St. Louis, MO, USA). To measure mRNA expression in A549-driven tumours, gastrocnemius muscle and epididymal white adipose tissue (eWAT), sacrificed tissues were frozen with liquid nitrogen and frozen tissues were pulverized by using dry ice and liquid nitrogen. Frozen tissues (30~40 mg) were homogenized with 1 mL of TRIzol (Invitrogen, Carlsbad, CA, USA) by using a bead homogenizer (Benchmark Scientific, Sayreville, NJ, USA) to extract total RNA. The purity of RNA was assessed by using a spectrophotometer (BioTek, Winooski, VT, USA) at 260 and 280 nm and a ratio of ~1.8 is accepted for measurement of gene expression. The reverse transcriptase and cDNA synthesis kit (Applied Biosystems, Foster City, CA, USA) was used for cDNA synthesis. SYBR Green qPCR mixture (Applied Biosystems, Foster City, CA, USA) was used for qRT-PCR and experimental Ct values were normalized and calculated to 36B4 (ribosomal protein subunit P0, RPLP0) or β-actin gene. Detailed primer sequences for qRT-PCR are described in Table 1.

### 2.2. Co-Immunoprecipitation and Western Blotting

For co-immunoprecipitation, total cell lysates were extracted by using 1% NP-40, 150 mM NaCl, 50 mM Tris-HCl (pH 7.9), 0.1 mM EDTA, and protease inhibitor cocktail containing lysis buffer. Cell lysates (2 mg/mL) were incubated with 20 μL of Flag-M2-affinity agarose gel (Sigma-Aldrich, St. Louis, MO, USA) for 16 h at 4 °C, then immunocomplexes were washed three times by using 0.5% NP-40, 200 mM NaCl, 50 mM Tris-HCl (pH 7.9), 0.1 mM EDTA and protease inhibitor cocktail containing wash buffer. For Western blotting, total proteins and eluted proteins after co-immunoprecipitation were subjected to SDS (sodium dodecyl sulfate)-PAGE (polyacrylamide gel electrophoresis), then proteins were transferred onto a PVDF membrane (Millipore, Burlington, MA, USA). Transferred proteins were reacted with primary antibodies for 16 h at 4 °C, and secondary antibodies for 1 h at room temperature, respectively. ECL Prime kit (GE Healthcare, Milwaukee, WI, USA) was used to visualize protein expression. Antibodies against TCF4 (H00006925-M04), β-actin (sc-47778), Flag-tag (F3165), Myc-tag (2278) and TWIST1 (ab50887) were purchased from Abnova (Taipei, Taiwan), Santa Cruz Biotechnology (Santa Cruz, CA, USA), Sigma-Aldrich (St. Louis, MO, USA), Cell Signaling Technology (Danvers, MA, USA) and Abcam (Burlingame, CA, USA), respectively.

### 2.3. Mammalian Two-Hybrid Luciferase Assay

pGL4.31-Luc (luc2P/GAL4UAS/Hygro) vector was used to measure VP16-TCF4 and GAL4-TWIST1 interaction-mediated luciferase expression. To perform high-throughput screening (HTS) by using 501 natural compounds, HEK293T cells were transiently transfected with pGL4.31-Luc, pFN10A-TCF4 and pFN11A-TWIST1, and then transfected cells were further incubated for 24 h to allow stabilization. Twenty-four hours post-transfection, cells were divided into 96-well cell culture plates and incubated for 24 h. Then, 5 μg/mL of natural compounds was treated in transfected cells and incubated for 24 h. To measure luciferase activities, cell lysates were reacted with luciferase assay buffer and luciferin substrate, and then luciferase activities were measured by using Luminometer (BioTek, Winooski, VT, USA).

### 2.4. Cell Culture and Transient Transfection

A549 (10185), NCI-H358 (25807) and Calu-1 (30054) lung cancer cells were obtained from Korean Cell Line Bank (Seoul, South Korea). Dulbecco’s modified Eagle’s medium (DMEM) and Roswell Park Memorial Institute (RPMI) 1640 with 10% fetal bovine serum (FBS) and antibiotics were used for cell culture. For transient transfection, mammalian expression vectors were transfected by using Polyfect (Qiagen, Hilden, Germany) in HEK293T, A549, Calu-1 and NCI-H358 cells. After transfection, cells were further incubated for 48 h to allow stabilization and protein expression.

### 2.5. Adenoviral Transduction, Mammalian Expression Plasmids and Gene Cloning

Adenovirus harbouring TCF4 (Gene Accession Number: BC031056) or GFP were purchased from Applied Biological Materials Inc (Richmond, BC, Canada). For adenoviral transduction, A549, Calu-1 and NCI-H358 lung cancer cells were incubated for 48 h with 1 × 10^4^ pfu/mL of adenovirus expressing GFP or TCF4. Myc-tagged TCF4 was provided from Dr. Jong-Wan Park at Seoul National University [30]. Mammalian expressing Flag-HA-TWIST1-FL (full length, aa 1–202), -NT1 (N-terminal 1, aa 1–105) and -NT2 (N-terminal 2, aa 1–165) are generated by using pLX304-V5-TWIST1 (GeneCopoeia, Rockville, MD, USA) and PCR containing primers with EcoR I and XhoI restriction site, and then amplified PCR fragments were inserted into pcDNA3.1-Flag-HA vector gift from Adam Antebi (Addgene plasmid # 52535). CheckMate^TM^/Flexi^®^ Vector Mammalian Two-Hybrid System

Kit (Promega, Madison, WI, USA) was used to generate a mammalian two-hybrid luciferase assay system. Full length (FL, aa 1–671), N-terminal domain (N, aa 1–250), middle domain (M, aa 251–500), and C-terminal domain (C, aa 501–671) of TCF4 was inserted into pFN10A vector expressing VP16-conjugated TCF4 proteins with *Sgf* I and *Pme* I restriction enzyme site. Full length of TWIST1 was inserted into pFN11A vector expressing GAL4-conjugated TWIST1 protein with *Sgf* I and *Pme* I restriction enzyme site. Detailed primer sequences for subcloning of TWIST1 domain mutants are described in Table 2.

### 2.6. Animal Experiment

The animal experiments were approved and performed in accordance with the guidelines of Konkuk University Institutional Animal Care and Committee (KU20062). A549 lung cancer cells (1 × 10^6^) suspensions were prepared in FBS and antibiotics-free RPMI1640: BD Matrigel^TM^ (BD Biosciences, Franklin Lakes, NJ, USA) mixture (100 µL) on ice and subcutaneously injected into the flank of Balb/c-nude mice. The mice were received normal animal diets (AIN93G) until a tumour size of 100 mm^3^ was reached. After the tumour size reached 100 mm^3^, the mice were divided into two groups and received daily doses of AIN93G or 2% Pc-Ex in their feed for 46 days. Pc-Ex-containing animal diets were formulated depending on nutrients of Pc-Ex such as carbohydrate, protein, lipid, and minerals in accordance with AIN93G (Table 3). The bodyweight in all of the mice was measured once a week. Tumour volumes were measured by using calipers once a week and calculated from the following equation: volume = ab^2^/2, where a is the maximal width and b is the maximal orthogonal width. When animal experiments were terminated, gastrocnemius muscle, eWAT and tumours were harvested, and all of the harvested tissues were frozen with liquid nitrogen.

### 2.7. ELISA for Measurement of PTHrP

LLC1 cells were seeded in a 12-well plate (1 × 10^4^ per well) and incubated with Pc-Ex (100 μg/mL) for 24 h in the presence of TGFβ1 (20 ng/mL). The supernatants of cells were collected and the PTHrP protein levels were measured using mouse ELISA (Enzyme-linked Immunosorbent Assay). All procedures were performed according to the manufacturer’s instructions (AVIVA SYSTEMS BIOLOGY, San Diego, CA, USA).

### 2.8. Preparation of Pc-Ex and Measurement of Emodin Concentration

For the preparation of *Polygonum cuspidatum* extracts, the powder of a dried *Polygonum cuspidatum* was obtained from the Jung-Ang microbe research institute (Cheongju, Korea). The powder was then immersed in water, sonicated for 15 min, and extracted for 24 h. The extract was filtered through non-fluorescent cotton and evaporated under reduced pressure by using a rotary evaporator at 40 °C. The condensed extract was then lyophilized by using a Modul Spin 40 dryer (Biotron Corporation, Calgary, AB, Canada) for 72 h. The plant extracts were stored at −20 °C and reconstructed with 50 mL of methanol before the HPLC analysis. All of the *Polygonum cuspidatum* extract samples were milled into powders, and an individual portion of the powdered samples (0.2 g) was dissolved in methanol. The resulting samples were then extracted for 15 min using an ultrasonic cleaner in a water bath (30 °C). The extracts were filtered by a GHP syringe filter and the elution was injected directly into the HPLC system for the analysis. Emodin (30269, ≥97) and formic acid (F0607, ≥95%) were purchased from Sigma-Aldrich (Sigma-Aldrich, St. Louis, MO, USA). Methanol, Water (HPLC-grade) was obtained from Fisher Scientific Co.(Fair Lawn, MA, USA). Double-distilled water was obtained using a Millipore Milli-Q Plus water treatment system (Millipore, Burlington, MA, USA). The stock solutions of five standards were made at a concentration of 10.3 mg in a final volume of 100 mL of methanol. Working solutions of mixed standards at the concentrations of 7.1, 14.1, 28.3, 56.5, and 113.0 μg were made by dilution of stock solution in volumetric flasks with the mobile phase. Then the standards were injected into the HPLC. High-performance liquid chromatography 20 μL samples were analyzed on a Zorbax Eclipse XDB-C18 column (4.6 × 250 mm, 5 μm, Agilent, Santa Clara, CA, USA), which was maintained at 40 °C using an Agilent Infinity-1260 HPLC system. The mobile phases consisted of (A) water containing 0.1% (*w*/*w*) formic acid and (B) methanol. The HPLC elution conditions were optimized as follows: linear gradient from 20 to 35% B (0 to 13 min), 35 to 100% B (13 to 30 min), and 100 to 20% B (30 to 40 min), where it was held for 3 min. The flow rate was set at 1.0 mL/min, and the column and autosampler were maintained at 30 and 25 °C, respectively. The scan range for the DAD detector system was set at 190 to 400 nm. Analogue output channel A was set at wavelength 254 nm with a bandwidth of 4 nm. The injection volume was 20 μL.

## 3. Results

### 3.1. PTHLH Expression Is Upregulated in Response to TGFβ1 in Lung Cancer

*PTHLH* is correlated with the pathogenesis of cancer cachexia, which is characterised by WAT browning, skeletal muscle loss, and bone metastasis [4,17]. In this study, the expression levels of *PTHLH* mRNA in patients with lung cancer were investigated using publicly available gene expression datasets. Bioinformatic analysis of GSE74706 [33] and GSE22863 datasets [34] revealed that the *PTHLH* mRNA levels were upregulated in non-small cell lung cancer (NSCLC) and primary lung cancer (Figure 1A). The TGFβ1 signalling pathway has a critical role in the biological processes of metazoans. The dysregulation of the TGFβ1 signalling pathway leads to tumour development and metastasis through the induction of EMT [35]. A recent study reported that the TGFβ1 signalling pathway upregulates the expression of *PTHLH* in lung and breast cancers [36]. Consistent with the findings of previous studies, the expression of *PTHLH* was significantly upregulated in TGFβ1-treated A549, NCI-H1299, NCI-H1650, NCI-H358, L132, Calu-1, and Calu-3 lung cancer cells (Figure 1B). Through bioinformatic analysis of the GSE114761 dataset [37] increased *PTHLH* mRNA levels were found in TGFβ1-treated many types of lung cancer cells (Figure 1C). These results indicate that *PTHLH* may be associated with TGFβ1-mediated lung cancer aggressiveness.

### 3.2. TCF4 Regulates PTHLH Expression in Lung Cancer Cells

Previous studies have reported that the TGFβ1 signalling pathway promotes the interaction between TCF4 and TWIST1 and consequently upregulates the expression of EMT-related genes [16]. The interaction between TCF4 and TWIST1 was also confirmed in this study (Figure 2A). TCF4 interacted with the bHLH domain of TWIST1 (Figure 2B). The expression of *PTHLH* is upregulated in TGFβ1-treated breast and lung cancer cells [17,38,39,40]. Thus, we hypothesized that TCF4 functions as an essential transcription factor to regulate *PTHLH* expression in response to TGFβ1 signalling. Adenoviral transduction of TCF4 upregulated *PTHLH* mRNA expression in A549, Calu-1, and NCI-H358 epithelial lung cancer cells (Figure 2C). The efficiency of adenoviral infection was examined by measuring the TCF4 protein levels (Figure 2D). The *PTHLH* mRNA levels were upregulated in lung cancer cells ectopically overexpressing Myc-TCF4 (Figure 2E). TGFβ1-induced *PTHLH* expression was mitigated in *TCF4*-silenced A549 (Figure 2F) and Calu-1 lung cancer cells (Figure 2G). These results indicate that TCF4 is involved in TGFβ1-induced *PTHLH* expression in lung cancer cells. To examine the functional role of TCF4 and TWIST1 in regulating *PTHLH* expression, the *PTHLH* mRNA levels were measured in lung cancer cells co-expressing TCF4 and TWIST1. The *PTHLH* and EMT-promoting *SNAI2* mRNA levels in the cells ectopically overexpressing TCF4 or TWIST1 alone was approximately twofold higher than those in cells infected and transfected with green fluorescence protein (GFP)-expressing adenovirus and empty vector (Figure 2H,I). Compared with those in the infected and transfected with GFP-expressing adenovirus and empty vector, the *PTHLH* and *SNAI2* mRNA levels were upregulated by approximately sevenfold (A549) or fourfold (NCI-H358) in cells co-expressing TCF4 and TWIST1 (Figure 2H,I). These results indicate that cooperation between TCF4 and TWIST1 promotes transcription of *PTHLH* and the expression of EMT-related genes, such as *SNAI2* in lung cancer cells.

### 3.3. Identification of a Natural Compound Targeting TCF4–TWIST1 Interaction Using Mammalian Two-Hybrid System-Based High-Throughput Screening (HTS)

The interaction between TCF4 and TWIST1 promotes lung cancer growth, metastasis, and cachexia by upregulating the expression of *PTHLH* and EMT-related genes. This indicated that this transcriptional axis is a potential druggable target for lung cancer growth, metastasis, and cachexia. The mammalian two-hybrid luciferase system is based on protein-protein interaction (PPI)-mediated luciferase expression. This system comprises GAL4-fusion and VP16-fusion proteins and is widely used to identify PPI inhibitors. GAL4-tagged TWIST1 and VP16-tagged N-terminal (N, amino acids 1–250), middle domain (M, amino acids 251–500), and C-terminal (C, amino acids 501–671) domain of TCF4 were generated (Figure 3A). The luciferase activities in the cells co-expressing GAL4-TWIST1 and full length (FL) and/or C-terminal (C) domain of VP16-TCF4 were higher than those in the cells expressing GAL4-TWIST1 alone (Figure 3A). To screen small molecules that target TCF4–TWIST1 interaction, HTS was performed using the mammalian two-hybrid system and 501 natural product compounds from a chemical library. Approximately 12 small molecules inhibited TCF4–TWIST1 interaction (Figure 3B). To exclude the possibility of downregulation of luciferase activities resulting from severe cell death, the cytotoxic activity of 12 natural compounds identified in Figure 3B was measured. As shown in Figure 3C, all compounds (5 µg/mL), except for emodin (5 µg/mL), exhibited potent cytotoxic activities. This indicated that emodin inhibits TCF4–TWIST1 interaction without exerting cytotoxic effects. Emodin dose-dependently mitigated the GAL4-TWIST1 and VP16-TCF4-FL-induced upregulation of luciferase activities (Figure 3D).

### 3.4. Emodin Inhibits TCF4/TWIST1 Complex-Induced PTHLH Expression

Next, we examined whether emodin was sufficient to dissociate the TCF4/TWIST1 complex. Treatment with emodin decreased the interaction between TCF4 and TWIST1 (Figure 4A). As shown in Figure 4B, emodin sufficiently inhibited the interaction between TCF4 and bHLH domain of TWIST1-NT2 (amino acids 1–165). These results indicate that emodin suppresses the interaction between TCF4 and TWIST1. As the TCF4/TWIST1 complex markedly upregulated *PTHLH* expression, the suppressive effect of emodin on TGFβ1-induced *PTHLH* and EMT-related gene expression was investigated. The expression of TGFβ1-induced *PTHLH* and EMT-promoting *CDH2* mRNAs was downregulated in emodin (20 µM)-treated NCI-H358 (Figure 4C) and A549 (Figure 4D) cells. This indicated that emodin exerts potential therapeutic effects on cachexia by downregulating the expression of *PTHLH* and EMT-related genes.

### 3.5. Emodin-Enriched Polygonum cuspidatum Extract (Pc-Ex) Inhibits TCF4–TWIST1 Interaction and Consequently Suppresses TGFβ1-Induced PTHLH Expression

Tumour-driven cytokines, such as growth differentiation factor 11, IL-1β, IL-6 leukaemiaia inhibitory factor, TNF-α, and TGFβ1 promote cancer cachexia with anorexia in the brain, proteolysis in the skeletal muscles, and browning in the WAT [5]. Thus, appetite-stimulating agents, such as megestrol (a synthetic derivative of naturally occurring progesterone) are widely used to treat cancer cachexia [41]. To develop a major component of medical nutrition therapy (MNT) for the treatment of lung cancer-induced cachexia, emodin-containing Pc-Ex was developed. The emodin content in Pc-Ex was confirmed using high-performance liquid chromatography (HPLC) (Figure 5A). Pc-Ex dose-dependently mitigated the GAL4-TWIST1 and VP16-TCF4-induced upregulation of luciferase activities (Figure 5B). Additionally, Pc-Ex sufficiently suppressed the interaction between TCF4 and TWIST1 (Figure 5C). Furthermore, Pc-Ex significantly mitigated the TGFβ1-induced upregulation of *PTHLH* mRNA expression in NCI-H358, A549, and Lewis Lung Carcinoma 1 (LLC1) cells (Figure 5D). Pc-Ex downregulated the PTHrP levels in LLC1 cells (Figure 5E). These results indicate that emodin-containing Pc-Ex is a potential agent for MNT-based treatment of cancer cachexia.

### 3.6. Pc-Ex Attenuates Lung Cancer-Induced Cachexia

Next, the pharmacological effect of Pc-Ex on lung cancer-induced cachexia was examined. Non-tumour-bearing (NTB) and A549 tumour-bearing BALB/c-nu mice were fed with a standard animal diet (AIN93G) or a feed supplemented with 2% Pc-Ex when the volume of the tumours reached approximately 100 mm^3^. The bodyweight (carcass weight) of A549 tumour-bearing mice was lower than that of NTB mice (Figure 6A). Additionally, the bodyweights of 2% Pc-Ex-administered A549 tumour-bearing mice were higher than those of the AIN93G-fed mice (Figure 6A). This indicated that emodin-containing Pc-Ex exerts therapeutic effects on body weight loss in tumour-bearing mice. Previous studies have reported that emodin exerts anti-cancer effects [42,43,44]. Consistently, the findings of this study indicated that the tumour growth in 2% Pc-Ex-administered mice was lower than that in the AIN93G-fed mice (Figure 6B). Additionally, A549 tumour-induced gastrocnemius muscle atrophy in 2% Pc-Ex-fed mice was alleviated when compared with that in AIN93G-fed A549 tumour-bearing mice (Figure 6C). Tumour-induced loss of gastrocnemius muscle weight was mitigated upon treatment with 2% Pc-Ex (Figure 6D). The *PTHLH* mRNA and PTHrP protein levels were downregulated in A549 tumours from 2% Pc-Ex-fed mice (Figure 6E,F). Next, the effect of emodin-containing Pc-Ex on the upregulated expression of white adipose (WAT) metabolism and skeletal muscle atrophy-related genes in tumour-bearing mice was examined. As shown in Figure 6G, the A549 tumour-induced expression of skeletal muscle atrophy-related genes, such as *Mstn*, *Fbxo32,* and *Trim63* was mitigated upon treatment with 2% Pc-Ex. Similarly, the administration of 2% Pc-Ex mitigated the A549 tumour-induced upregulation of WAT metabolism-related gene (*Pgc1a and Acox1*) in epididymal WAT (Figure 6H). These results indicate that emodin-containing Pc-Ex can mitigate the wasting of fat and skeletal muscle in lung cancer-induced cachectic mice.

## 4. Discussion

PTHrP, an essential paracrine and autocrine ligand encoded by PTHLH, regulates bone homeostasis and the initiation, growth, and progression of various types of cancer [14]. The upregulated expression of oncogenic PTHLH is associated with poor prognosis owing to its role in promoting tumour initiation, growth, angiogenesis, metastasis, and chemoresistance in pancreatic [19], colorectal [18], intrahepatic cholangiocarcinoma [45], head and neck [46], osteosarcoma [47], and breast cancers [17,48]. In this study, the correlation between PTHLH expression and LUAD and LUSC was examined using bioinformatics analysis. The analysis of TCGA datasets revealed that the PTHLH mRNA levels in the LUAD and LUSC tissues were upregulated when compared with those in the non-tumorous lung tissues. Additionally, the upregulated PTHLH mRNA levels were correlated with decreased survival in patients with LUAD. These results indicate that PTHLH is a potential diagnostic biomarker and a molecular therapeutic target for lung cancer.

Lung cancer-derived PTHrP promotes the wasting of WAT and skeletal muscles, which results in cancer cachexia [4]. However, the precise molecular mechanisms underlying the production and secretion of PTHrP in lung cancer cells are poorly understood. TGFβ1, a well-known secreted protein from the tumour, immune, and tumour-associated fibroblast cells, promotes tumour cell migration, invasion, and metastasis [49]. Additionally, TGFβ1 functions as a critical stimulus for promoting cancer cachexia, which is characterised by severe bodyweight loss resulting from skeletal muscle loss and WAT browning [7]. Based on these findings, we hypothesized that TGFβ1-primed tumour cells exhibit PTHrP expression in lung cancer cells. Consistent with the findings of previous studies, this study demonstrated that the expression of *PTHLH* was upregulated in response to TGFβ1 signalling in various lung cancer cell lines [38,39,40]. This suggests that *PTHLH* is a critical component for TGFβ1-induced cachexia in lung cancer.

TCF4 (also known as ITF2 and E2-2), a class I bHLH transcription factor, functions as a transcriptional activator or a transcriptional repressor depending on its binding partners [24]. Previous studies have reported that TCF4 regulates cancer development and progression [25,26,29,50,51]. Additionally, TCF4 is reported to regulate the transcription of pro-metastatic and EMT-related genes in malignant melanoma and lung cancer [24,26]. TGFβ1 signalling upregulates TCF4 expression in cancer cells [16,52]. However, the clinical roles of TCF4 in cancer cachexia have not been previously investigated. Thus, we hypothesized that TCF4 could function as an upstream transcription factor for TGFβ1 signalling-induced *PTHLH* expression. The *PTHLH* mRNA levels were upregulated upon TCF4 overexpression and downregulated upon *TCF4* silencing in lung cancer cells. These results indicate that TCF4 is an essential transcription factor that regulates TGFβ1-mediated *PTHLH* expression.

TCF4 can interact with multiple types of transcription factor and function as a transcriptional activator or a transcriptional repressor depending on its binding partners during various biological processes, such as cell cycle, proliferation, differentiation, and development [17]. The interaction between Tcf4 and Math1 regulates neuronal progenitor differentiation and is associated with the pathogenesis of mental disorders [53]. TWIST1, a bHLH transcription factor, plays an essential role during embryonic development through the regulation of EMT-related genes [31]. Previous studies have demonstrated that TCF4 can interact with TWIST1 to promote the expression of EMT-related genes, such as *CDH2*, *VIM*, *SNAI1*, and *SNAI2*, in embryonic stem and lung cancer cells [16,32]. This study demonstrated that TCF4 strongly binds to the bHLH domain of TWIST1 to form a heterodimer. The interaction between TCF4 and TWIST1 was confirmed using the mammalian two-hybrid interaction assay. TCF4–TWIST1 interaction regulates TGFβ1 signalling-induced EMT and cancer metastasis [14]. However, the functional role of the TCF4/TWIST1 complex in PTHrP-associated cancer cachexia has not been previously investigated. In this study, the *PTHLH* mRNA levels in A549 and NCI-H358 cells co-expressing TCF4 and TWIST1 were higher than those in cells expressing TCF4 or TWIST1 alone. These results suggest that TCF4/TWIST1 complex-induced *PTHLH* expression may promote cachexia in lung cancer and that these molecular frameworks are promising therapeutic targets for cancer cachexia.

*Polygonum cuspidatum* (also known as *Reynoutria japonica*) is a traditional Chinese medicinal herb with a wide range of pharmacological activities, including anti-inflammatory, anti-cancer, and anti-hyperlipidaemic activities [54]. Various active compounds, including emodin (quinone), resveratrol (stilbene), and quercetin (flavonoid), have been isolated from the *P. cuspidatum*. These phytochemicals are reported to exert beneficial effects on various chronic diseases [54]. Emodin (1,3,8-trihydroxy-6-methylanthraquinone) exerts multiple pharmacological effects, including anti-allergic, anti-osteoporotic, anti-diabetic, neuroprotective, hepatoprotective, cardioprotective, and anti-cancer effects [55,56], on various chronic diseases. A pre-clinical pharmacokinetic study has demonstrated the safety of emodin in mice [57]. Treatment with low (20 mg/kg body weight), medium (40 mg/kg body weight), or high (80 mg/kg body weight) doses of emodin for 12 weeks did not affect the physiology of major organs, such as the kidneys and liver. This indicated the potential application of emodin, a natural ingredient, for the treatment of human chronic diseases [57]. Emodin is reported to suppress pancreatic cancer-induced cachexia [58]. Additionally, emodin inhibited tumour-induced hepatic gluconeogenesis and skeletal muscle proteolysis in MiaPaca2 cell-implanted athymic mice [58]. Although emodin is reported to inhibit TGFβ1 signalling [59,60,61,62], the suppressive effects of emodin on TGFβ1-associated cancer cachexia have not been previously investigated. In this study, Pc-Ex, which contains emodin as a major component, sufficiently suppressed the *PTHLH* mRNA and PTHrP expression levels in lung cancer cells stimulated with TGFβ1 by inhibiting the interaction between TCF4 and TWIST1. Additionally, the administration of Pc-Ex downregulated *PTHLH* expression in A549 tumour-bearing mice and significantly decreased tumour growth, increased skeletal muscle weight, reversed the expression of proteolysis and fat browning-related genes in the skeletal muscle and WAT, respectively, in cachectic mice. These findings demonstrate that emodin-enriched Pc-Ex is a potential therapeutic for cancer cachexia.

Mice were fed on an AIN93G diet supplemented with 2% Pc-Ex in the form of edible pellets for a month. The daily treatment dose of Pc-Ex was 2400 mg/kg body weight for each mouse. HPLC analysis of emodin contents in Pc-Ex indicated that the dietary emodin treatment dose was 12 mg/kg body weight. Previous studies have revealed the hepatotoxic effects of emodin [57]. Although Pc-Ex-formulated diets comprise low amounts of emodin, mouse hepatic metabolomic profile must be examined for the application of Pc-Ex to suppress tumour growth and cachexia. The contents and half-life of emodin in the blood of mice administered with Pc-Ex can be analysed using pharmacokinetic analysis, which can aid in determining the pharmacological potential of emodin for cancer-induced cachexia.

Mechanistically, the expression of *PTHLH* in multiple cancers is regulated by various transcription factors and signalling cascades. The inactivation or downregulation of p38MAPK signalling is reported to suppress *PTHLH* expression in lung and liver metastases of primary colorectal cancer [63]. Smad3, a transcription factor, upregulates the transcription of *PTHLH* by binding to the proximal *PTHLH* promoter region in response to TGFβ1 signalling in breast cancer cells and consequently promotes bone metastasis [17]. The Runx2-mediated activation of *PTHLH* expression promotes the growth of head and neck squamous cell carcinoma [46]. In this study, the interaction between TCF4 and TWIST1 upregulated *PTHLH* expression and consequently promoted lung cancer growth and cachexia. These findings revealed the mechanistic framework involved in the regulation of *PTHLH* expression, which can aid in the development of novel therapeutic strategies for lung cancer metastasis and lung cancer-induced cachexia.

## 5. Conclusions

The interaction between TCF4 and TWIST1 upregulated *PTHLH* expression in lung cancer cells in response to TGFβ1 stimulation. This indicated that this transcriptional framework may be involved in the pathogenesis of lung cancer-induced cachexia. The major findings of this study are as follows: TCF4 promoted TWIST1-mediated *PTHLH* expression; emodin and Pc-Ex suppressed TGFβ1-induced *PTHLH* expression by inhibiting the interaction between TCF4 and TWIST1; emodin-containing Pc-Ex alleviated skeletal muscle wasting and fat browning and downregulated the expression of atrophy-associated genes in tumour-bearing mice. Thus, the findings of this study indicated that the interaction between TCF4 and TWIST1 is a potential molecular therapeutic target for lung cancer-induced cachexia and that emodin and emodin-enriched Pc-Ex, which inhibit TCF4–TWIST1 interaction, are promising therapeutic candidates for lung cancer-induced cachexia.

## Figures and Tables

**Figure 1 nutrients-14-01508-f001:**
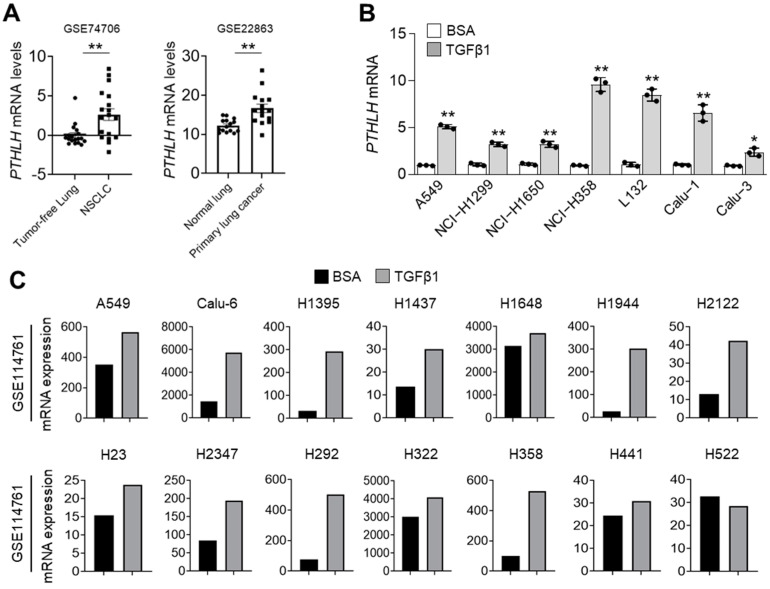
TGFβ1 increases PTHLH expression in lung cancer cells. (**A**) The mRNA expressions of PTHLH in human normal lung and lung cancer tissues were evaluated by using RNA Seq V2 RSEM. The box plots reveal the distribution of values. All datasets were downloaded from cBioportal on 9 November 2021 (www.cbioportal.org). (**B**) PTHLH mRNA levels were measured by using qRT-PCR. All lung cancer cells were incubated with BSA or TGFβ1 (20 ng/mL) for 24 h. (**C**) The mRNA expressions of PTHLH in TGFβ1-treated lung cancer cells were analyzed by using RNA Seq V2 RSEM. Dataset was downloaded from cBioportal. All plots indicate the mean ± SD. * *p* < 0.05 and ** *p* < 0.01 by Student’s *t*-test for two experimental comparisons.

**Figure 2 nutrients-14-01508-f002:**
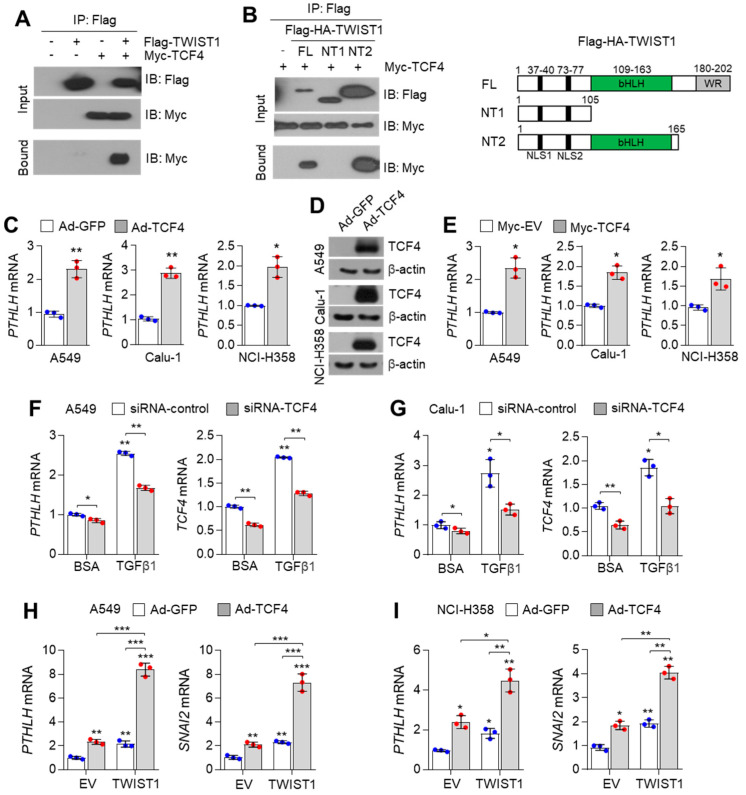
TCF4 increases PTHLH expression in lung cancer cells. (**A**) Interaction of TCF4 and TWIST1. Myc-tagged TCF4 and Flag-tagged TWIST1 were transfected into HEK293T cells, then cells were incubated for 48 h. Interaction of TCF4 and TWIST1 was checked by co-immunoprecipitation assay and Western blotting. (**B**) The basic helix–loop–helix (bHLH) domain of TWIST1 is required for interaction with TCF4. Myc-TCF4, Flag-TWIST1-FL (full length, aa 1–202), -NT1 (N-terminal 1, aa 1–105) and -NT2 (N-terminal 2, aa 1–165) expressing plasmids were transfected into HEK293T cells as indicated. At 48 h post-transfection, the interaction of TCF4 and TWIST1 was detected by co-immunoprecipitation assay and Western blotting. (**C**,**D**) Transduction of adenovirus harbouring human TCF4 increases PTHLH expression. Cells were infected with adenovirus harboring GFP (1 × 10^4^ pfu/mL) or TCF4 (1 × 10^4^ pfu/mL), and then incubated for 48 h. (**C**) PTHLH mRNA levels were measured by using qRT-PCR and (**D**) overexpressed TCF4 protein was confirmed by using Western blotting. (**E**) Ectopic expression of TCF4 increases PTHLH expression. Myc-tagged TCF4 or empty vector were transiently transfected into A549, Calu-1 or NCI-H358 cells, and then cells were further incubated for 48 h. PTHLH mRNA levels were measured by using qRT-PCR. (**F**,**G**) TCF4 knock-down decreases PTHLH in TGFβ1-treated lung cancer cells. Control or TCF4 targeting siRNA (20 nM) were transfected into (**F**) A549 and (**G**) Calu-1 cells, then transfected cells were incubated for 48 h in the absence or presence of TGFβ1 (20 ng/mL). TCF4 knock-down efficiency was checked by measuring TCF4 mRNA levels. PTHLH mRNA levels were measured by using qRT-PCR. (**H**,**I**) Co-expression of TCF4 and TWIST1 enhance PTHLH and SNAI2 expression. A549 (**H**) and NCI-H358 (**I**) cells were transiently transfected with Flag-TWIST1. 12 h post-transfection, cells were infected with adenovirus harbouring GFP or TCF4, then cells were further incubated for 48 h. PTHLH and SNAI2 mRNA levels were measured by using qRT-PCR. The values represent the mean ± SD of three independent experiments performed in duplicate; * *p* < 0.05, ** *p* < 0.01 and *** *p* < 0.001. Statistical analysis was performed using Student’s *t*-test (**C**,**E**) and one-way ANOVA Tukey post hoc test (**F**–**I**). Blue points: raw data of control group, Red points: raw data of experimental group.

**Figure 3 nutrients-14-01508-f003:**
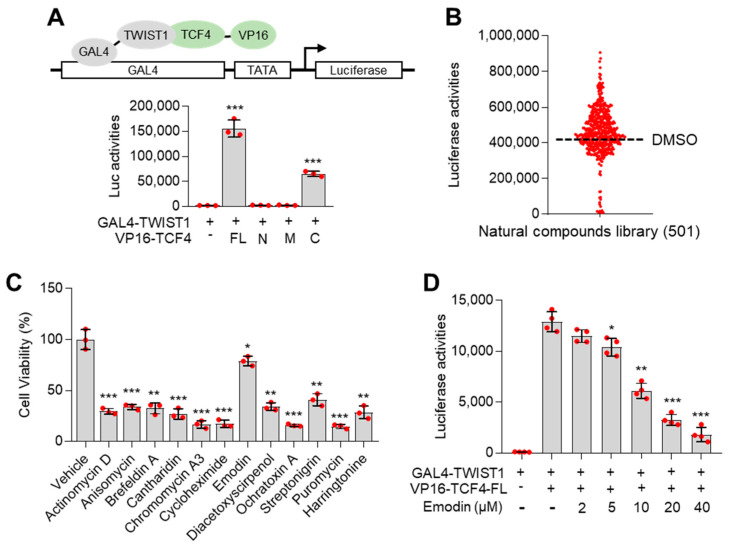
Mammalian two-hybrid system-based high-throughput screening (HTS) for identification of natural compounds inhibiting TCF4–TWIST1. (Administrator) **(A)** Confirmation of interaction of TCF4 and TWIST1 by using a mammalian two-hybrid system. Principle of the mammalian two-hybrid system (upper panel). HEK293T cells were transiently transfected with GAL4-TWIST1, VP16-TCF4-full length (FL), VP16-TCF4-N-terminal domain (N), -middle domain (M) and C-terminal domain (C) expressing plasmids with pGL4.31-luciferase vector, and then cells were further incubated for 48 h. Luciferase activities were analyzed by using luciferase assay. (**B**) High-throughput screening for identification of natural compounds inhibiting the interaction of TCF4 and TWIST1. HEK293T cells were transfected with GAL4-TWIST1, VP16-TCF4-full length (FL) and pGL4.31-luciferase vector, and then transfected cells were incubated for 48 h prior to treatment of natural compounds library. Post-transfection, cells were incubated with 5 µg/mL of the natural compound for 24 h. Luciferase activities were analyzed by using luciferase assay. (**C**) Measurement of cell viability in hit compounds-treated cells. HEK293T cells were incubated for 24 h with each hit compound (5 µg/mL) as indicated. Cell viability was measured by crystal violet staining and assay. (**D**) Emodin is a potential small molecule targeting the interaction of TCF4 and TWIST1. HEK293T cells transfected with GAL4-TWIST1, VP16-TCF4-full length (FL) and pGL4.31-luciferase plasmids were incubated for 24 h with emodin as indicated, then luciferase activities were analyzed. The values represent the mean ± SD of three independent experiments performed in duplicate; * *p* < 0.05, ** *p* < 0.01 and *** *p* < 0.001. Statistical analysis was performed by using a one-way ANOVA Tukey post hoc test.

**Figure 4 nutrients-14-01508-f004:**
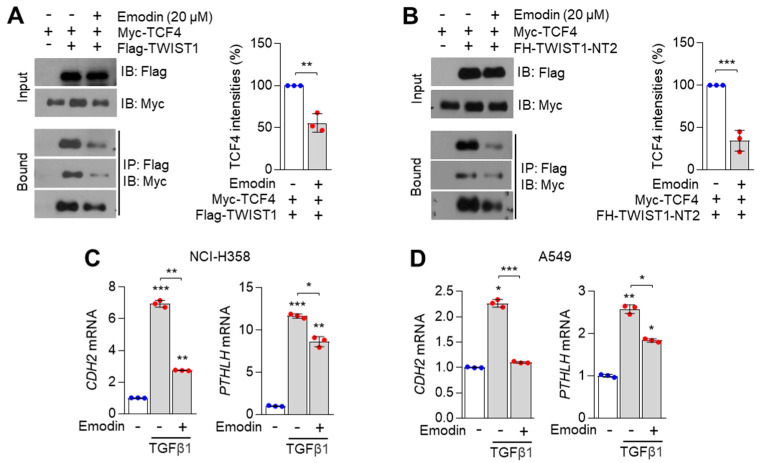
Emodin inhibits the cooperation of TCF4 and TWIST1 on PTHLH expression. (**A**,**B**) Emodin dissociates the interaction of TCF4 and TWIST1. HEK293T cells were transiently transfected with Myc-TCF4 and (**A**) Flag-TWIST1 or (**B**) Flag-TWIST1-NT2, then cells were incubated for 24 h. Post-transfection, cells were further incubated for 12 h in the absence or presence of 20 μM of emodin. Interaction of TCF4 and TWIST1 was measured by using co-immunoprecipitation assay and Western blotting. (**C**,**D**) Emodin suppresses TGFβ1-induced PTHLH expression in NCI-H358 and A549 lung cancer cells. (**C**) NCI-H358 and (**D**) A549 cells were incubated with TGFβ1 (20 ng/mL) for 1 h prior to emodin treatment, followed by further incubation with or without emodin (20 μM) for 24 h. CDH2 and PTHLH mRNA levels were measured by qRT-PCR. The values represent the mean ± SD of three independent experiments performed in duplicate; * *p* < 0.05, ** *p* < 0.01 and *** *p* < 0.001. Statistical analysis was performed by using a one-way ANOVA Tukey post hoc test.

**Figure 5 nutrients-14-01508-f005:**
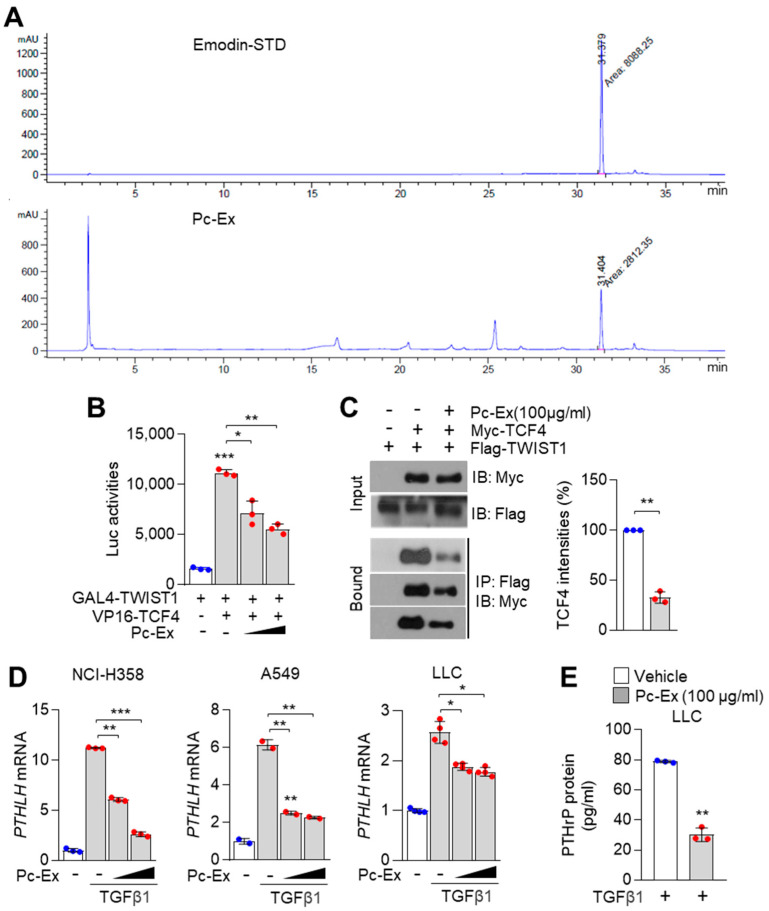
Emodin-containing polygonum cuspidatum extract (Pc-Ex) suppresses PTHrP expression with decreased interaction of TCF4 and TWIST1. (**A**) The quantitative concentration of emodin in *Polygonum cuspidatum* extract (Pc-Ex). UPLC was used to measure the quantitation of emodin concentration in Pc-Ex. (**B**) Pc-Ex suppresses the interaction of TCF4 and TWIST1. HEK293T cells were transfected with GAL4-TWIST1, VP16-TCF4-FL and pGL4.31-luciferase vector, and then transfected cells were incubated for 24 h prior to treatment of Pc-Ex. Post-transfection, cells were incubated with Pc-Ex (50 and 100 µg/mL) for 24 h. Luciferase activities were analyzed by using luciferase assay. (**C**) Pc-Ex dissociates the interaction of TCF4 and TWIST1. HEK293T cells were transiently transfected with Myc-TCF4 and Flag-TWIST1, then cells were incubated for 24 h. Post-transfection, cells were further incubated for 24 h in the absence or presence of 100 µg/mL of Pc-Ex. Interaction of TCF4 and TWIST1 was measured by using co-immunoprecipitation assay and Western blotting. (**D**) Pc-Ex attenuates TGFβ1-induced PTHLH expression in NCI-H358, A549, and LLC1 lung cancer cells. Cells were incubated with TGFβ1 (20 ng/mL) for 1 h prior to Pc-Ex treatment, followed by further incubation with or without Pc-Ex (50 and 100 µg/mL) for 24 h. PTHLH mRNA levels were measured by qRT-PCR. The values represent the mean ± SD of three independent experiments performed in duplicate; * *p* < 0.05, ** *p* < 0.01 and *** *p* < 0.001. Statistical analysis was performed by using a one-way ANOVA Tukey post hoc test. (**E**) Pc-Ex decreases PTHrP upon TGFβ1. LLC1 cells were incubated for 48 h in the absence or presence of 100 µg/mL of Pc-Ex with 20 ng/mL of TGFβ1. Mouse PTHrP protein levels were measured by using ELISA assay. The values represent the mean ± SD of three independent experiments performed in duplicate; ** *p* < 0.01. Statistical analysis was performed by using the unpaired Student’s *t*-test.

**Figure 6 nutrients-14-01508-f006:**
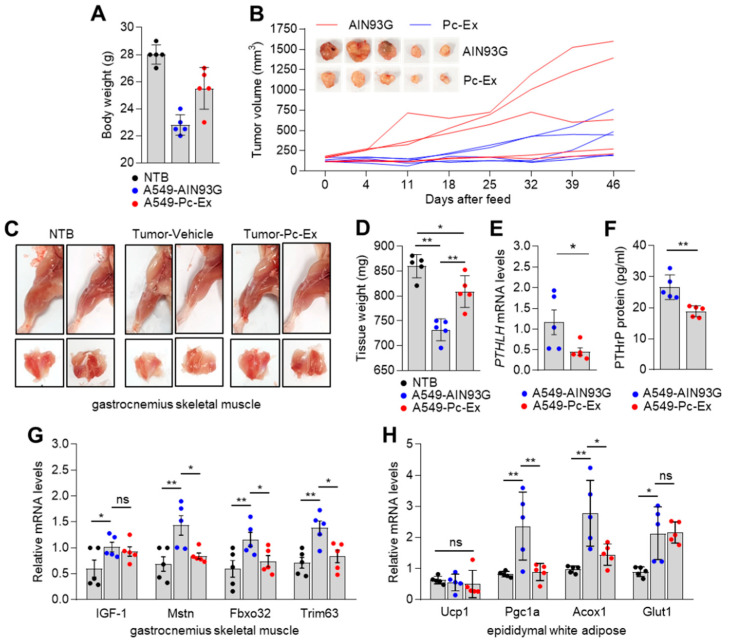
Pc-Ex suppresses tumour growth and alleviates A549 tumour-driven cachexia. (**A**) Body weight (carcass weight) of mice. Non-tumour-bearing (NTB) or A549 tumour-bearing mice were received AIN93G or 2% Pc-Ex every day for 46 days after tumour size reached 100 mm^3^ (*n* = 5). (**B**) Administration of Pc-Ex decreases tumour growth. Representative images of isolated tumours and growth curves of every single tumour are shown. Mice inoculated with A549 cells were sacrificed after daily administration of AIN93G or 2% Pc-Ex. (**C**) Representative images of isolated hindlimb and gastrocnemius muscles are shown. (**D**) Administration of Pc-Ex increases muscle weight in A549 tumour-bearing mice. (**E**) Pc-Ex suppresses PTHLH mRNA and (**F**) PTHrP protein expression in A549-driven tumour. (**G**) Pc-Ex reverses muscle atrophy-and (**H**) epididymal white adipose (eWAT) metabolism-related genes expression in gastrocnemius muscle and eWAT of A549 tumour-bearing mice. mRNA levels in gastrocnemius muscle and eWAT were determined by RT-qPCR. The values represent the mean ± SEM. * *p* < 0.05 and ** *p* < 0.01. Statistical analysis was performed by using the one-way ANOVA Tukey post hoc test.

**Table 1 nutrients-14-01508-t001:** Primer sequences for qRT-PCR.

Gene	Forward Sequences (5′-3′)	Reverse Sequences (5′-3′)
**CDH2 (h)**	CCACCTTAAAATCTGCAGGC	GTGCATGAAGGACAGCCTCT
**SNAI2 (h)**	TGACCTGTCTGCAAATGCTC	CAGACCCTGGTTGCTTCAA
**TCF4 (h)**	CATAGGGAGTCCCATCTCCA	GGACCAACTTCTTTGGCAAG
**PTHLH (h)**	TTGTCATGGAGGAGCTGATG	CGGTGTTCCTGCTGAGCTAC
**36B4 (h)**	TGGTGATACCTAAAGCCTGGAA	CATGTTGCTGGCCAATAAGG
**Mstn (m)**	CAGGAGAAGATGGGCTGAATC	AGTGCTCATCGCAGTCAAG
**Fbxo32 (m)**	ACCCAAGAAGAGAGCAGTATG	GACTCCCAGCCATCCAATTA
**Trim63 (m)**	CAAGGAACACGAAGACGAGAA	TCCTCCAGCTGAGAGATGAT
**IGF1 (m)**	GTCGTCTTCACACCTCTTCTAC	CTCATCCACAATGCCTGTCT
**UCP1 (m)**	ACAGAAGGATTGCCGAAACT	CTGTAGGCTGCCCAATGAA
**PGC1α (m)**	AAACTGACTTCGAGCTGTACTT	CCCATGAGGTATTGACCATCTC
**Acox1 (m)**	TGCCTTTGTTGTCCCTATCC	GTCCATCTTCAGGTAGCCATTAT
**Glut1 (m)**	TCTGTCGGCCTCTTTGTTAATC	CCAGTTTGGAGAAGCCCATAA
**B** **-actin (m)**	ACGAGGCCCAGAGCAAGAG	TCTCCAAGTCGTCCCAGTTG

**Table 2 nutrients-14-01508-t002:** Primer sequences for cloning.

Constructs	Sequences (5′-3′)
**Flag-HA-TWIST1-FL**	[F] CCGCTCGAGATGATGCAGGACGTG
	[R] CCGGAATTCTTAGTGGGACGCGGACAT
**Flag-HA-TWIST1-NT1**	[F] CCGCTCGAGATGATGCAGGACGTG
	[R] CCGGAATTCTTACTCCTCGTAAGACTG
**Flag-HA-TWIST1-NT2**	[F] CCGCTCGAGATGATGCAGGACGTG
	[R] CCGGAATTCTTAGCTCTGGAGGACCTG
**pFN10A-TCF4-FL**	[F] ATAGGCGATCGCCATGCATCACCAACAGCGA
	[R] AGCTTTGTTTAAACCATCTGTCCCATGTGATTCGATGC
**pFN10A-TCF4-N**	[F] AGCTTTGTTTAAACTCCTGGTGGCATGCCTCT
	[R] AGCTTTGTTTAAACCTGTGGAATATGAGAAGAGTTGCCC
**pFN10A-TCF4-M**	[F] ATAGGCGATCGCCATGTCCAGCAGCTACTGTAGCC
	[R] AGCTTTGTTTAAACTCCTGGTGGCATGCCTCT
**pFN10A-TCF4-C**	[F] ATAGGCGATCGCCATGCTACAGGGGCAGAGTGT
	[R] AGCTTTGTTTAAACCATCTGTCCCATGTGATTCGATGC
**pFN11A-TWIST1**	[F] ATAGGCGATCGCCATGATGCAGGACGTGTCCA
	[R] AGCTTTGTTTAAACGTGGGACGCGGACATGGA

**Table 3 nutrients-14-01508-t003:** Nutrition contents in AIN93G and 2% Pc-Ex feeds.

Nutrition	AIN93G (g)	2% Pc-Ex (g)
**Casein**	200	194.38
**Corn starch**	397.486	383.34
**Dextrose**	132	132
**Sucrose**	100	100
**Cellulose**	50	50
**Soybean Oil**	70	69.76
**t-Butylhydroquinone**	0.014	0.014
**Salt Mix**	35	35
**Vitamin Mix**	10	10
**L-cystine**	3	3
**Choline Bitartrate**	2.5	2.5
**Pc-Ex**	0	20
**Total**	1000	1000

## Data Availability

All materials generated in this study are available from the corresponding author on reasonable request.

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
