# Peer review of "Polygonum cuspidatum Extract (Pc-Ex) Containing Emodin Suppresses Lung Cancer-Induced Cachexia by Suppressing TCF4/TWIST1 Complex-Induced PTHrP Expression"

_nutrients, 2022, doi:10.3390/nu14071508_

Round 1

Reviewer 1 Report

In the study entitled: “Polygonum cuspidatum root extract (Pc-Ex) containing emodin 2 suppresses lung cancer-induced cachexia by suppressing 3 TCF4/TWIST1 complex-induced PTHrP expression” Fang and colleagues showed the molecular mechanisms underlying the TCF4/TWIST1 interaction-mediated PTHrP expression in lung cancer-induced cachexia. Overall, the study is well conducted, the results are clearly reported and the conclusion supports the findings. Nevertheless, few points should be considered to further improve the quality of the study.

Major points:

  • Lines 445-447: “the body weight of 2% Pc-Ex-administered A549 tumour-bearing mice was higher than that of the AIN93G-fed mice (Figure 6A)”. According with the data shown in the figure however, this trend does not reach a significant difference. I suggest representing the body weight of figure 6A as carcass-weight, defined as the total weight minus the tumor weight.
  • The results of the study of tumor-related muscle adaptations shown in figure 6 could benefit of morphometric analyses of muscle tissue. Please, include the analyses of fiber area and physiological cross sectional area for each experimental condition of Figure 6.
  • Because of the crucial role of Atrogin1 in promoting cancer-related muscle fiber atrophy the analysis of its expression should be included in the list of muscle atrophy-related genes of figure 6G.

Minor points:

  • Please, spell out the acronyms listed in the abstract and in the text at least once (PTHLH, PTHrP, TCF4…)
  • Because of the role of PTHLH 477 as potential diagnostic biomarker and a molecular therapeutic target for lung cancer, as also highlighted by the authors, the introduction should include few lines about the state of the art of the biomarkers in cancer cachexia (doi: 10.3390/diagnostics11010116, doi: 10.3389/fcell.2021.720096. eCollection 2021)

Author Response

POINT-BY-POINT RESPONSE TO THE REVIEWER 1

We would like to thank the reviewers for the comments as well as the time expend on our manuscript. We have now addressed all the previous points with additions to the text and specific explanations. Additionally, we have found typos and corrected them: “Polygonum cuspidatum root extract” was corrected as Polygonum cuspidatum extract.

In the study entitled: “Polygonum cuspidatum root extract (Pc-Ex) containing emodin suppresses lung cancer-induced cachexia by suppressing  TCF4/TWIST1 complex-induced PTHrP expression” Fang and colleagues showed the molecular mechanisms underlying the TCF4/TWIST1 interaction-mediated PTHrP expression in lung cancer-induced cachexia. Overall, the study is well conducted, the results are clearly reported and the conclusion supports the findings. Nevertheless, few points should be considered to further improve the quality of the study.

We totally agree with the reviewer’s comment and thank the reviewer to give us positive evaluation and constructive comments.

Major points:

Lines 445-447: “the body weight of 2% Pc-Ex-administered A549 tumor-bearin mice was higher than that of the AIN93G-fed mice (Figure 6A)”. According with the data shown in the figure however, this trend does not reach a significant difference. I suggest representing the body weight of figure 6A as carcass-weight, defined as the total weight minus the tumor weight.

We thank the reviewer to give us a critical point. The body weight in Figure 6A means carcass weight. To exclude the reader’s misunderstanding, we have commented that body weight meant carcass weight in the manuscript (Line 448) and figure legends (Figure 6A).

The results of the study of tumor-related muscle adaptations shown in figure 6 could benefit of morphometric analyses of muscle tissue. Please, include the analyses of fiber area and physiological cross sectional area for each experimental condition of Figure 6.

We really apologize for not performing the reviewer’s suggestion, because we did not achieve the sacrificed muscle tissues for morphometric analyses. Although we cannot show atrophic fiber, we would like to prove the effects of emodin-enriched Pc-Ex on cancer-induced cachexia by muscle-atrophy markers determined by qPCR and muscle photos presented in Figure 6.

Because of the crucial role of Atrogin1 in promoting cancer-related muscle fiber atrophy the analysis of its expression should be included in the list of muscle atrophy-related genes of figure 6G.

We appreciate the reviewer’s comment. Atrogin 1 is also known as Fbxo32 (F-box only protein 32), and we determined the mRNA levels of Fbxo32 in the muscle of mice (Figure 6G). We have addressed the sentence regarding the alternative name of Fbxo32 (Line 55).

Minor points:

Please, spell out the acronyms listed in the abstract and in the text at least once (PTHLH, PTHrP, TCF4…)

According to the reviewer’s comment, we now have spelled out the acronyms such as PTHLH, PTHrP, and TCF4 in the abstract section.

Because of the role of PTHLH 477 as potential diagnostic biomarker and a molecular therapeutic target for lung cancer, as also highlighted by the authors, the introduction should include few lines about the state of the art of the biomarkers in cancer cachexia (doi: 10.3390/diagnostics11010116, doi: 10.3389/fcell.2021.720096. eCollection 2021)

Thanks to the reviewer’s comment, we have now described the importance of cachexia-related biomarkers in the introduction section (Line 72-76). Then the manuscript would strengthen the rationale of PTHLH as cancer cachexia biomarkers. Furthermore, we have added related references (Ref 12-13)

Reviewer 2 Report

In general, the study is complex but the paper is easy to follow having a logical flow. The results are novel and original, providing an advance in the field. Moreover, considering the possibility to improve life quality in lung cancer-associated cachexia the results are important. Emodin and emodin-enriched Pc-Ex can be future targeted personalized therapies.

Author Response

POINT-BY-POINT RESPONSE TO THE REVIEWER 2

We would like to thank the reviewers for the comments as well as the time expend on our manuscript. We have now addressed all the previous points with additions to the text and specific explanations. Additionally, we have found typos and corrected them: “Polygonum cuspidatum root extract” was corrected as Polygonum cuspidatum extract.

In general, the study is complex but the paper is easy to follow having a logical flow. The results are novel and original, providing an advance in the field. Moreover, considering the possibility to improve life quality in lung cancer-associated cachexia the results are important. Emodin and emodin-enriched Pc-Ex can be future targeted personalized therapies.

We thank the reviewer to give us positive evaluation.